# Novel Immobilized Biocatalysts Based on Cysteine Proteases Bound to 2-(4-Acetamido-2-sulfanilamide) Chitosan and Research on Their Structural Features

**DOI:** 10.3390/polym14153223

**Published:** 2022-08-08

**Authors:** Svetlana S. Olshannikova, Nataliya V. Malykhina, Maria S. Lavlinskaya, Andrey V. Sorokin, Nikolay E. Yudin, Yulia M. Vyshkvorkina, Anatoliy N. Lukin, Marina G. Holyavka, Valeriy G. Artyukhov

**Affiliations:** 1Biophysics and Biotechnology Department, Voronezh State University, 394018 Voronezh, Russia; 2Metagenomics and Food Biotechnologies Laboratory, Voronezh State University of Engineering Technologies, 394036 Voronezh, Russia; 3Polymer Science and Colloid Chemistry Department, Voronezh State University, 394018 Voronezh, Russia; 4Phystech School of Biological and Medical Physics, Moscow Institute of Physics and Technology, 141701 Dolgoprudnyi, Russia; 5Research Core Center of Voronezh State University, Voronezh State University, 394018 Voronezh, Russia; 6Physics Department, Sevastopol State University, 299053 Sevastopol, Russia

**Keywords:** cysteine proteases, ficin, papain, bromelain, 2-(4-acetamido-2-sulfanilamide) chitosan, complexation

## Abstract

Briefly, 2-(4-Acetamido-2-sulfanilamide) chitosan, which is a chitosan water-soluble derivative, with molecular weights of 200, 350, and 600 kDa, was successfully synthesized. The immobilization of ficin, papain, and bromelain was carried out by complexation with these polymers. The interaction mechanism of 2-(4-acetamido-2-sulfanilamide) chitosan with bromelain, ficin, and papain was studied using FTIR spectroscopy. It was found that the hydroxy, thionyl, and amino groups of 2-(4-acetamido-2-sulfanilamide) chitosan were involved in the complexation process. Molecular docking research showed that all amino acid residues of the active site of papain formed hydrogen bonds with the immobilization matrix, while only two catalytically valuable amino acid residues took part in the H-bond formation for bromelain and ficin. The spectral and in silico data were in good agreement with the catalytic activity evaluation data. Immobilized papain was more active compared to the other immobilized proteases. Moreover, the total and specific proteolytic activity of papain immobilized on the carrier with a molecular weight of 350 kDa were higher compared to the native one due to the hyperactivation. The optimal ratio of protein content (mg × g ^−1^ of carrier), total activity (U × mL^−1^ of solution), and specific activity (U × mg^−1^ of protein) was determined for the enzymes immobilized on 2-(4-acetamido-2-sulfanilamide) chitosan with a molecular weight of 350 kDa.

## 1. Introduction

The research and development of new antibacterial substances is an important goal for modern chemistry, biology, and medicine. Due to the wide availability of some antibiotics and their uncontrolled use, many pathogenic microorganisms have developed resistance to their action. This significantly reduces the effectiveness of therapy according to currently relevant treatment protocols [1,2,3,4,5]. Moreover, often-used antibacterial drugs are characterized by non-selective activity against pathogens. Treatment with such drugs leads to pronounced side effects caused by a significant increase, even to toxic values, in their concentration in the blood. This process, also known as burst release, is followed by a decrease to a level below therapeutic values in the concentration of the drug [6,7]. All these factors necessitate the search for new biologically active substances with antibacterial properties, as well as the creation of new pharmaceutical forms of existing drugs to rationalize their practical use and prolong the action of a biologically active substance.

Cysteine proteases are a group of enzymes catalyzing the hydrolysis of proteins and peptides. They are widely distributed in nature and found in both animals and plants. Some of them, such as plant bromelain (EC 3.4.22.32/3.4.22.33), papain (EC 3.4.22.2), and ficin (EC 3.4.22.3), have an antibacterial effect. The mechanism of their action is the following: the enzymes cause the decomposition of proteins loosening the cell wall, leading to cell leakage, swelling, the destruction of the bacterial membrane, and cell damage. Bromelain, papain, and ficin are widely used for various medical purposes, such as wound dressing, as well as the removal of necrotic and infected tissues in wounds or burns [8,9,10,11].

Bromelain, ficin, and papain are globular proteins consisting of two clearly differentiated domains which are α-helix (*L*-domain) and a β-barrel-like structure (*R*-domain) [12]. Their active site is located at the interface of the *L*- and *R*-domains and is formed by cysteine, histidine, and aspartic acid residues. The cysteine and histidine residues form an ion catalytic pair stabilized by aspartic acid fixing the histidine imidazole cycle. The cysteine thiol group interacts with the carbonyl group of a substrate peptide bond, forming a tetrahedral intermediate. The one collapses to regenerate the carbonyl group, resulting in an acyl-enzyme complex, which is hydrolyzed into the free enzyme and the *N*-terminal portion of the product [12].

It is known that the dissolved forms of enzymes are characterized by low stability, which limits their use in drug development. This problem can be solved by immobilizing the enzyme on an inert or antibacterial matrix [13,14,15,16,17,18]. Immobilization increases the rigidity of the tertiary structure of the enzyme molecules, thereby stabilizing it and increasing the half-life [19].

The promising matrices for enzymes immobilization that have attracted research attention in recent decades are carbon nanostructured materials (such as carbon (functionalized) nanotubes, graphene or its oxide, etc.) [20,21], and natural polysaccharides, as well as their derivatives, particularly chitosan, which is characterized by antibacterial activity [22]. Interestingly, biocompatibility and non-toxicity mean that chitosan and its derivatives are potential candidates for both conventional and novel drug delivery systems. Moreover, chitosan-based matrixes are of interest in tissue engineering for controlled drug release, as well as for tissue remodeling, due to their fibrous and porous properties [23,24,25].

Ficin [8], papain [26], and bromelain [27] have already been immobilized using chitosan, but hyperactivation effects were not observed. Ficin and papain immobilized on chitosan have excellent antimicrobial and anti-biofilm properties against *Staphylococci* microorganisms [8,18]. Moreover, immobilized papain and ficin are capable of batch desorption, which makes them less cytotoxic compared to native enzymes [18].

To expand the possibilities of using chitosan, it is advisable to carry out its chemical modification. It is well known that macromolecules of polysaccharides are bound by a large number of intra- and intermolecular hydrogen bonds formed between the functional groups of the pyranose cycles [28]. This affects their ability to dissolve in water: chitosan with molecular weight > 10,000 is soluble only in acidic media with pH < 6.5 [29]. The introduction of functional groups that block hydrogen bond formation centers makes it possible to increase the pH range of the solubility of this polymer. In addition, the introduction of new groups can increase conjugation ability or biological activity. For example, the introduction of succinic residues into macromolecules acid enhances the antibacterial properties of chitosan [30].

Therefore, we propose for the first time to use water-soluble derivatives of chitosan–2-(4-acetamido-2-sulfanilamide) chitosan for the immobilization of some cysteine proteases (ficin, papain, bromelain). This polymer is characterized by antifungal activity [31]. Moreover, it contains moieties of sulfanilamide which is an antibacterial agent; so, 2-(4-acetamido-2-sulfanilamide) chitosan can be characterized by higher activity compared to the unmodified chitosan. Thus, the synergistic effects of combining the antimicrobial properties of cysteine proteases and 2-(4-acetamido-2-sulfanilamide) chitosan can be expected.

This work aims to develop biocatalysts based on cysteine proteases in complex with 2-(4-acetamido-2-sulfanilamide) chitosan and study their structural features.

## 2. Materials and Methods

### 2.1. Materials

Bromelain (B4882), papain (P4762), and ficin (F4165) were purchased from Sigma-Aldrich, Germany and were used without any treatments. Azocasein (Sigma-Aldrich, Munich, Germany) was used as a hydrolysis substrate in catalytic activity evaluation experiments. Chitosan with molecular weights of 200, 350, and 600 kDa and a degree of deacetylation of 0.73–0.85 (Bioprogress, Shchelkovo, Russia), 4-acetylsulfanilyl chloride (Sigma-Aldrich, Munich, Germany), dimethyl sulfoxide, and acetone (both analysis grade, ReaKhim, Moscow, Russia) were applied for 2-(4-acetamido-2-sulfanilamide) chitosan synthesis.

### 2.2. 2-(4-Acetamido-2-sulfanilamide) Chitosan Synthesis and Characterization

For a typical experiment, 2 g of chitosan and 150 mL of dimethyl sulfoxide (DMSO) were placed in a thermostatically controlled reactor equipped with a stirrer and a reflux condenser, and the mixture was dispersed for 30 min at room temperature. Then, a solution of 4-acetylsulfanilyl chloride in 15 mL of DMSO was added dropwise (chitosan: reagent = 2:3 mol/mol) and the mixture was kept at 60 °C with constant stirring for 6 h. The resulting product was isolated by precipitation into acetone, the precipitate was filtered off and washed several times with acetone, after which the products were dried in a vacuum oven to constant weight. The product yields were 75–83%.

The chitosan modification was confirmed by FTIR spectroscopy on a Bruker Vertex-70 spectrometer (Bremen, Germany). The degrees of substitution of the products obtained were calculated from the FTIR data by correlating the areas under the absorption bands at 1035 cm^−1^ corresponding to the skeletal vibrations of the chitosan pyranose cycles and at 1402 cm^−1^ related to the vibrations of the thionyl groups. The obtained values were 0.327, 0.286, and 0.167 for the polymers with molecular weights of 200, 350, and 600 kDa, respectively.

### 2.3. Molecular Docking

The preparation of the structure of the cysteine proteases for docking [19] and the process of modeling interactions were carried out as described by Holyavka et al. [32,33]. The PDB IDs for enzyme input files were the following: 1W0Q for bromelain, 4YYW for ficin, and 9PAP for papain.

The enzyme structures were prepared for docking according to the standard scheme for the Autodock Vina package (The Scripps Research Institute, San-Diego, CA, USA): the atoms (together with their coordinates) of the solvent, buffer, and ligand molecules were removed from the PDB input file. Before carrying out the numerical calculations, a charge was placed on the surface of the proteins using the MGLTools. The center of the molecule and the parameters of the box («cells») were set manually, ensuring that the whole protease molecule could fit in the box.

The 2-(4-acetamido-2-sulfanilamide) chitosan structure model was drawn in the HyperChem molecular designer (HyperCube Inc., Waterloo, ON, Canada); this structure was consistently optimized first in the AMBER force field, and then quantum-chemically in PM3. The ligands in the docking calculations had the maximum conformational freedom: the rotation of the functional groups around all single bonds was allowed. Charge arrangement on the chitosan molecule and its protonation/deprotonation was performed automatically in the MGLTools 1.5.6 package (The Scripps Research Institute, San-Diego, CA, USA).

### 2.4. Enzyme Immobilization

Enzyme immobilization was carried out by complexation. Firstly, to 1 g of 2-(4-acetamido-2-sulfanilamide) chitosan was added 20 mL of an enzyme solution (1 mg × mL^−1^ in 0.05 M glycine buffer with a pH of 10.0 for ficin; 0.05 M glycine buffer with a pH of 9.0 for papain; and 0.05 M Tris-glycine buffer with a pH of 9.0 for bromelain) and incubated for 2 h at 37 °C. After that, the formed precipitate was washed by dialysis against 50 mM Tris-HCl buffer with a pH of 7.5 through a cellophane membrane with a 25 kDa pore size until the absence of protein in the washing water (control was carried out on an SF-2000 spectrophotometer (LOMO-microsystems, Saint Petersburg, Russia) at λ = 280 nm).

### 2.5. Protein Content Measurement

The protein content in the immobilized enzymes was determined by the modified Lowry method [34]. The modification consisted of the decomposition of the bonds between the carrier matrix and the enzyme molecule in the first step of the analysis. The immobilized enzyme was treated with the solution of K, Na-tartrate (in a concentration of 20 mg × mL^−1^ or 0.7 M) that was prepared from 1 M NaOH at 50 °C for 10 min. Earlier, proteins were shown to be completely desorbed under these conditions [17,35]. The absence of the enzyme destruction processes was controlled by recording its absorption spectrum on a UV-2550PC spectrophotometer (Shimadzu Scientific Instruments Inc., Kyoto, Japan).

### 2.6. Proteolytic Activity Evaluation of the Immobilized Enzymes

Azocasein was chosen as the hydrolysis substrate for proteolytic activity determination [36]. The experiments were conducted in 50 mM Tris-HCl buffer with pH 7.5 at 37 °C as described in [32]. The substrate concentration was 0.4% w. Briefly, the sample was dissolved in 200 μL of buffer (50 mM Tris-HCl, pH 7.5), mixed with 800 μL of azocasein solution (0.5% in the same buffer), and incubated for 30 min at the temperatures indicated above. Then, 800 μL of 5% trichloroacetic acid (TCA) solution was added, and after 10 min incubation at 4 °C, the precipitated unhydrolyzed azocasein was removed by centrifugation (3 min 13,000 rpm). The supernatant (1200 μL) was mixed with 240 μL of 1 M NaOH solution, and its optical density was measured at 410 nm. The unit of catalytic activity was taken as the amount of enzyme that hydrolyzed 1 μM of the substrate in 1 min (Μm × min^−1^ × mg^−1^).

## 3. Results

Figure 1 represents the FTIR spectra of 2-(4-acetamido-2-sulfanilamide) chitosan (Figure 1) and its complexes with bromelain, ficin, and papain. The FTIR spectrum of 2-(4-acetamido-2-sulfanilamide) chitosan contained the following characteristic absorption bands described in Table 1. These bands were observed in the spectra of the polymer with enzymes; however, some changes indicated the formation of the complex. The C-O-C bonds of the pyranose cycles were reflected as a two-component band with two maxima at 1035 and 1050 cm^−1^ in the chitosan derivative spectrum, and the intensity of the component at 1050 cm^−1^ was lower compared to the one at 1035 cm^−1^. In the spectra of the enzyme complexes, the intensity of these components became more similar. This demonstrated the involvement of the OH-groups of 2-(4-acetamido-2-sulfanilamide) chitosan in the enzyme complex formation. Moreover, the wavenumber shifts were observed for some bands. The wavenumber increased for ν_as_(SO_2_), ν_breath_(benzene cycles), and δ_plan_(N-H), and it decreased for δ_plan_(C-H). These patterns were present in the spectra of all researched enzymes. More interesting changes occurred with the Amide I bands. Its intensity decreased for all complexes; however, the bromelain and ficin wavenumber decreased, as the one did not shift for the papain complex. This indirectly indicated a significant change in the conformations of bromelain and ficin, while papain underwent fewer conformation changes.

An analysis of the protein content in the immobilized enzymes showed that the largest amount of ficin and bromelain (mg × g^−1^ carrier) bound to a polymer with a molecular weight of 600 kDa, and there was no significant effect of the molecular weight on the papain content (Figure 2A). The total activity of ficin (U × mL^−1^ of solution) was higher in the complex with 2-(4-acetamido-2-sulfanilamide) chitosan with a molecular weight of 600 kDa, while the one for papain and bromelain was higher for the complexes with polymers with a molecular weight of 350 kDa (Figure 2B). All immobilized enzymes in complex with 2-(4-acetamido-2-sulfanilamide) chitosan with a molecular weight of 350 kDa showed the highest specific activity (U × mg^−1^ of protein) (Figure 2C). This was probably due to the polymer matrix with a molecular weight of 350 kDa providing the best steric accessibility of the active site for the substrate.

Moreover, papain immobilized on 2-(4-acetamido-2-sulfanilamide) chitosan with a molecular weight of 350 kDa had a higher proteolytic activity compared to the native one. The increase in enzymatic activity compared to the free enzyme may be due to several mechanisms. First, the activity of papain can increase after immobilization due to a change in the conformation of its molecule, which has a positive effect on the reaction rate. Secondly, enzymes can be inhibited by high concentrations of the substrate or some reaction products, which reduces the observed activity, while immobilization completely or partially prevents inhibition. [42]. In addition, in the case of papain, immobilization probably suppresses such inactivation processes as aggregation and autolysis. The fact is that in the immobilized state, the mobility of proteins is, as a rule, sharply limited, while the essential stage of both mechanisms is the movement of two (or several) protease molecules to each other with their subsequent interaction. [43]. Considering the efficiency of complexation for this case, it is clear that hyperactivation occurred. Moreover, the results obtained correlated with the FTIR and molecular docking data.

For a more detailed explanation of the results obtained, we performed in silico experiments on the molecular docking (Figure 3, Table 2). Interestingly, the percentage of the retention activity of bromelain and ficin after complexation was comparable; however, there was a significantly different number of hydrogen bonds and physical interactions formed by them with the 2-(4-acetamido-2-sulfanilamide) chitosan matrix. The highest percentage of remaining activity was observed for papain. This is probably due to the entire catalytic triad being involved in the process of complexation: Cys25 and Asp158 form hydrogen bonds, and His159 generates van der Waals interactions with the matrix. For the other enzymes, only two amino acid residues in the active site participate in the process of complexation: Cys23 and His158 form hydrogen bonds in bromelain, while Cys22 and Asp161 form hydrogen bonds in ficin.

## 4. Conclusions

In the present work, we synthesized the water-soluble chitosan derivatives, 2-(4-acetamido-2-sulfanilamide) chitosan with molecular weights of 200, 350, and 600 kDa. For the first time, the cysteine proteases ficin, papain, and bromelain, were immobilized by complexation with these polymers. The investigation of the interaction mechanism between the component researched by FTIR and molecular docking shows that all catalytically valuable amino acid residues of bromelain and ficin participated in the hydrogen bond formations, while papain generated two hydrogen bonds through the amino acid residues of the active site. This feature in cooperative action with the influence of the steric factors of 2-(4-acetamido-2-sulfanilamide) chitosan with a molecular weight of 350 kDa leads to papain hyperactivation and an increase in proteolytic activity. At the same time, the catalytic activity of the other immobilized enzymes was lower compared to the native ones. The optimal ratio of protein content (mg × g^−1^ of carrier), total activity (U × mL^−1^ of solution), and specific activity (U × mg^−1^ of protein) was determined for the complexes of all the studied proteases with 2-(4-acetamido-2-sulfanilamide) chitosan with a molecular weight of 350 kDa. We expect complexes of 2-(4-acetamido-2-sulfanilamide) with cysteine proteases—ficin, bromelain, and especially papain appears to be a beneficial agent for outer wound treatment capable of the biofilm’s destruction, increasing the efficacy of wound treatment.

## Data Availability

Not applicable.

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
