# Peer review of "Novel Immobilized Biocatalysts Based on Cysteine Proteases Bound to 2-(4-Acetamido-2-sulfanilamide) Chitosan and Research on Their Structural Features"

_polymers, 2022, doi:10.3390/polym14153223_

Round 1
Reviewer 1 Report
This article presented the synthesis and characterization of 2-(4-Acetamido-2-sulfanilamide) chitosan, and its interaction with bromelain, ficin, and papain. Protease papain immobilized on 2-(4-acetamido-2-sulfanimide) chitosan with a molecular weight of 350 kDa was successfully achieved. This research topic is well filled within the scope of the journal. However, this paper needs suitable revision before acceptance.
1. The abstract section could highlight the results of papain immobilization with detailed data.
2. Line 120, Tris-glycine buffer pH 9.0 was used? Please check it carefully.
3. Section 2.5, what are the conditions of pH and temperature used for the activity assay?
4. For the molecular docking study, the PDB numbers should be noted.
5. What’s the protein concentration of the enzyme (bromelain, ficin, and papain) used for the experiment of immobilization?
Author Response
Reviewer #1
Comment 1: The abstract section could highlight the results of papain immobilization with detailed data.
Response 1: The abstract section was corrected according to your recommendation. All corrections are highlighted in yellow.
Comment 2: Line 120, Tris-glycine buffer pH 9.0 was used? Please check it carefully
Response 2: Yes, it is correct. The bromelain immobilization was carried out in 0.05 M Tris-glycine buffer with a pH of 9.0
Comment 3: Section 2.5, what are the conditions of pH and temperature used for the activity assay?
Response 3: The activity assays were performed in 50 mM Tris-HCl buffer with pH 7.5 at 37 °C. This and other information were added to the 2.6 Subsection.
Comment 4: For the molecular docking study, the PDB numbers should be noted.
Response 4: PDB IDs were added to 2.3 Subsection
Comment 5: What’s the protein concentration of the enzyme (bromelain, ficin, and papain) used for the experiment of immobilization?
Response 5: The information on the enzymes’ concentration was added to the 2.4 Subsection
Thank you for your comments! You help us become better!
Reviewer 2 Report
This paper deals with bioactive compounds featuring antibacterial properties in a context where they are anchored onto a derivatized form of the natural polymeric biomolecule chitosan. The field of research is highly commendable as it offers new opportunities in biomedicine and could eventually become valuable addressing the challenges with antibiotics resistance. Not questioning the validity of the results presented and te conclusions drans from them by the research team, the way in which the work and the conclusions are displayed and summarized in the manuscript leave a lot to be desired. Thus, the authors are recommended to conduct a major overhaul of their paper turning the contents into a much more significant and convincing story, which would give it a better accessibility to the readers in the scientific community.
Alongside, the authors also have to improve the English language both in the sense of sharpening the way in which many of the sentences are expressed as well as avoiding to comit to so many typos. The authors are recommended to be exert more care in their proof-reading and leave the sloppy standard demonstrated here behind.

Author Response
Reviewer #2
Comment: This paper deals with bioactive compounds featuring antibacterial properties in a context where they are anchored onto a derivatized form of the natural polymeric biomolecule chitosan. The field of research is highly commendable as it offers new opportunities in biomedicine and could eventually become valuable addressing the challenges with antibiotics resistance. Not questioning the validity of the results presented and te conclusions drans from them by the research team, the way in which the work and the conclusions are displayed and summarized in the manuscript leave a lot to be desired. Thus, the authors are recommended to conduct a major overhaul of their paper turning the contents into a much more significant and convincing story, which would give it a better accessibility to the readers in the scientific community.
Alongside, the authors also have to improve the English language both in the sense of sharpening the way in which many of the sentences are expressed as well as avoiding to comit to so many typos. The authors are recommended to be exert more care in their proof-reading and leave the sloppy standard demonstrated here behind.
Response: We try to re-represent our research in a more readable and accurate form. Also, English was proofread. Hope you will find the new representation of our work suitable for publication in the Polymers journal.
Thank you for your comments! You help us become better!
Reviewer 3 Report
Dear Authors,
the topic that you develop in the manuscript is interesting. However, the descriptions You made are too general and the text must be extended to include essential details. In my opinion, this is necessary to consider your work for publication in . I recommend a major revision of the manuscript before the decision on further publication steps can be made.
Overall comments to the manuscript that should be taken under consideration by the authors
(1) Section 1. Introduction (Page 1-2) should definitely be improved. In the current state, it is too general. The main suggestion is to point out if the 2-(4-acetamido-2-sulfanilamide) chitosan has been used for the first time in this study? Or there were previous literature reports on this topic?
Moreover, the main goals of the study and the potential of such immobilized preparation (protease – chitosan derivative) should be more emphasized.
In addition, I suggest putting 2-3 sentences about the synergistic effect of combining the antimicrobial properties of a chitosan-based matrix and the catalytic ability of tested enzymes to decompose the bacterial cell wall proteins.
(2) The Section 2. Materials and Methods (Page 2-3) and Section 3. Results (Page 3-8) should be revised and completed with some essential information (see detailed remarks to the manuscript).
(3) The Section 4. Conclusions (Page 10, lines 191 – 200) are not actually the summary of obtained results. Instead, the authors provide only a shortened list of all the experiments performed and methods used in the presented study. This cannot be accepted in its present form.
Authors are asked to give a comparison of their results with those obtained by other research groups. Moreover, they are asked to give some information on how the immobilized biocatalysts they received could be applied in medicine as antibacterial systems. The future perspective of this research direction should be described as well.
(4) Authors are asked to check the spelling of the word ‘2-(4-acetamido-2-sulfanilamide)’ in the whole text because there are mistakes (e.g. line 3, 27, 38, etc.)
(5) The manuscript requires a precise English proofreading
Detailed remarks to the manuscript
(1) Page 1, Title – I suggest to improve the title. In the current form, there is not clear.
My proposition: ‘Novel immobilized biocatalysts based on cysteine proteases bound to 2-(4-acetamido-2-sulfanimide) chitosan and research of their structure features’
(2) Page 1, line 34 – I suggest to change ‘…was found for immobilized enzymes on -(4-acetamido-2-sulfanilamide) chitosan …’ to ‘…was determined for enzymes immobilized on -(4-acetamido-2-sulfanimide) chitosan …’
(3) Page 1, lines 35 – 37 – ‘The combination of relatively cheap components reveals the availability of the proposed technological method for laboratories and gives prospects for further use of the immobilized enzymes in biomedicine.’
This statement is unclear and confusing. As a main goal of the study, the Authors give: ‘to develop biocatalysts based on cysteine proteases in complex with 2-(4-acetamido-2-sulfanilamide) chitosan and study their structural features’. Based on this I guess that the work is not focused on providing any technological method, but mainly on the features of obtained immobilized enzymes. Please clarify and revise this statement.
(4) Page 2, line 72 – I would like to ask if the authors are sure that they mean ‘substrate’ here or rather the ‘product’?
(5) Page 2, line 83 – Please change the word ‘substance’ to ‘compound’ or ‘agent’.
(6) Page 3, Subsection 2.3 – It is necessary to add a brief description of the procedure used for molecular docking. The reference only to the source article is not enough.
(7) Page 4, Subsection 2.4, lines 124 – 125 – Please explain how you make the Lowry test for immobilized enzymes? Or you determined the protein content for immobilized proteases from the difference between the overall amount of proteins in the soluble enzyme preparation taken to the immobilization and the amount of protein unbound to the chitosan-based matrix.
(8) Page 3, Subsection 2.5 – Authors are asked to give a brief description of the conditions used for testing the catalytic activity of enzymes (e.g. reaction temperature and time, the volume of substrate and enzyme, wavelength, etc.)
(9) Page 7, Fig. 2 – Authors are asked to explain how papain after immobilization on chitosan support (350 kDa) may possess 102% of specific activity (Fig.2c)/ 122% of total activity (Fig.2b) in comparison with the soluble enzyme. Whereas the yield of protein immobilization is only 60% (Fig.2a)? If you assumed that it is the result of occurring the phenomenon of hyperactivation after immobilization or any other, it must be described in the text.

Author Response
Reviewer #3
Overall comments:
Comment 1: Section 1. Introduction (Page 1-2) should definitely be improved. In the current state, it is too general. The main suggestion is to point out if the 2-(4-acetamido-2-sulfanilamide) chitosan has been used for the first time in this study? Or there were previous literature reports on this topic? Moreover, the main goals of the study and the potential of such immobilized preparation (protease – chitosan derivative) should be more emphasized. In addition, I suggest putting 2-3 sentences about the synergistic effect of combining the antimicrobial properties of a chitosan-based matrix and the catalytic ability of tested enzymes to decompose the bacterial cell wall proteins.
Comment 3: The Section 4. Conclusions (Page 10, lines 191 – 200) are not actually the summary of obtained results. Instead, the authors provide only a shortened list of all the experiments performed and methods used in the presented study. This cannot be accepted in its present form. Authors are asked to give a comparison of their results with those obtained by other research groups. Moreover, they are asked to give some information on how the immobilized biocatalysts they received could be applied in medicine as antibacterial systems. The future perspective of this research direction should be described as well.
Response 1 and 3: We try to rewrite the Introduction section to make it less general. As well as try to rewrite the Conclusion section (3rd comments).
Comment 2: The Section 2. Materials and Methods (Page 2-3) and Section 3. Results (Page 3-8) should be revised and completed with some essential information (see detailed remarks to the manuscript).
Response 2: The required corrections were added to the 2 and 3 Sections.
Comment 4: Authors are asked to check the spelling of the word ‘2-(4-acetamido-2-sulfanilamide)’ in the whole text because there are mistakes (e.g. line 3, 27, 38, etc.)
Response 4: The carrier’s name was checked and standardized in the manuscript.
Comment 5: The manuscript requires a precise English proofreading
Response 5: Proofreading was also done.
Detailed remarks:
Comment 1: Page 1, Title – I suggest to improve the title. In the current form, there is not clear. My proposition: ‘Novel immobilized biocatalysts based on cysteine proteases bound to 2-(4-acetamido-2-sulfanimide) chitosan and research of their structure features’
Response 1: The title was corrected according to your suggestion. All corrections are highlighted in yellow.
Comment 2: Page 1, line 34 – I suggest to change ‘…was found for immobilized enzymes on -(4-acetamido-2-sulfanilamide) chitosan …’ to ‘…was determined for enzymes immobilized on -(4-acetamido-2-sulfanimide) chitosan …’
Response 2: The required change was performed.
Comment 3: Page 1, lines 35 – 37 – ‘The combination of relatively cheap components reveals the availability of the proposed technological method for laboratories and gives prospects for further use of the immobilized enzymes in biomedicine. This statement is unclear and confusing. As a main goal of the study, the Authors give: ‘to develop biocatalysts based on cysteine proteases in complex with 2-(4-acetamido-2-sulfanilamide) chitosan and study their structural features’. Based on this I guess that the work is not focused on providing any technological method, but mainly on the features of obtained immobilized enzymes. Please clarify and revise this statement.
Response 3: The unclear and confusing statement was deleted from the Abstract.
Comment 4: Page 2, line 72 – I would like to ask if the authors are sure that they mean ‘substrate’ here or rather the ‘product’?
Response 4: Of course, the term ‘product’ means in this sentence.
Comment 5: Page 2, line 83 – Please change the word ‘substance’ to ‘compound’ or ‘agent’.
Response 5: The required change was performed
Comment 6: Page 3, Subsection 2.3 – It is necessary to add a brief description of the procedure used for molecular docking. The reference only to the source article is not enough.
Response 6: A brief description of the procedure used for molecular docking was added to 2.3 Subsection
Comment 7: Page 4, Subsection 2.4, lines 124 – 125 – Please explain how you make the Lowry test for immobilized enzymes? Or you determined the protein content for immobilized proteases from the difference between the overall amount of proteins in the soluble enzyme preparation taken to the immobilization and the amount of protein unbound to the chitosan-based matrix.
Response 7: The information on protein content measurement by the modified Lowry method is added to the 2.5 Subsection
Comment 8: Page 3, Subsection 2.5 – Authors are asked to give a brief description of the conditions used for testing the catalytic activity of enzymes (e.g. reaction temperature and time, the volume of substrate and enzyme, wavelength, etc.)
Response 8: The activity assays were performed in 50 mM Tris-HCl buffer with pH 7.5 at 37 °C. This and other information were added to the 2.6 Subsection.
Comment 9: Page 7, Fig. 2 – Authors are asked to explain how papain after immobilization on chitosan support (350 kDa) may possess 102% of specific activity (Fig.2c)/ 122% of total activity (Fig.2b) in comparison with the soluble enzyme. Whereas the yield of protein immobilization is only 60% (Fig.2a)? If you assumed that it is the result of occurring the phenomenon of hyperactivation after immobilization or any other, it must be described in the text.
Response 9: The explanation of the catalytic activity increase of immobilized papain was added on page 5, lines 205-208.
Thank you for your comments! You help us become better!
Reviewer 4 Report
The authors present an interesting describing the synthesis of 2-(4-Acetamido-2-sulfanimide) chitosan with different molecular weights along with the immobilization of ficin, papain, and bromelain. The rationale for conducting this work is well-justified and the paper aligns with the scopes of the journal.
There are some points that need to be clarified and some additional characterization is essential in order to be published.
1. Please write in a more detailed paragraph the potential applications of these structures. These materials can be successfully immobilized in carbon materials. You should add a paragragh in the introduction for enhancing the novelty of your work. Some references that can help you:
doi:10.1021/acsami.2c03944. (Graphene Oxide–Cytochrome c Multilayered Structures for Biocatalytic Applications: Decrypting the Role of Surfactant in Langmuir–Schaefer Layer Deposition).
doi: 10.1016/bs.mie.2019.10.015 (Use of functionalized carbon nanotubes for the development of robust nanobiocatalysts).
2. Please include XPS measurements revealing the bonds that are present.
3. Please include XRDs patterns.
Author Response
Reviewer # 4
Comment 1: Please write in a more detailed paragraph the potential applications of these structures. These materials can be successfully immobilized in carbon materials. You should add a paragragh in the introduction for enhancing the novelty of your work. Some references that can help you: doi:10.1021/acsami.2c03944. (Graphene Oxide–Cytochrome c Multilayered Structures for Biocatalytic Applications: Decrypting the Role of Surfactant in Langmuir–Schaefer Layer Deposition). doi: 10.1016/bs.mie.2019.10.015 (Use of functionalized carbon nanotubes for the development of robust nanobiocatalysts).
Response 1: The recommended paragraph about carbon nanomaterials used in immobilization was added to the Introduction part (page 2, lines 77-92)
Comment 2 and 3: Please include XPS measurements revealing the bonds that are present. Please include XRDs patterns.
Response: Undoubtedly, the use of the XPS and XRD methods could allow us to look at our objects from the other side and expand our understanding of the interactions between the matrix and enzymes. XPS method makes it possible to analyze the physicochemical state of a surface layer no more than a few nm thick. However, in our opinion, the application of the XPS method to systems like ours is extremely painstaking and time-consuming work that requires updating instrument calibrations, and time-consuming sample preparation. For a more “bulky state” analysis of a sample by XPS, it is necessary to carry out its profiling by etching with ions or argon ion clusters. This is currently a non-trivial task in application to (bio)organic objects, which requires the development of a methodology for interpreting experimental data, but profiling by argon ion treatment leads to the destruction of the surface of the studied sample and the loss of valuable experimental data. Undoubtedly, the application of the XPS method to bioorganic samples is extremely promising nowadays and will expand the horizons of representations of the "structure-property" relationship. Therefore, they will be the subject of our further research because of the high complexity and limited time provided for correcting current work.
The XRD method makes it possible to obtain volumetric characteristics of samples with a sufficiently high degree of ordering and/or nanostructuring. In the case of sufficiently amorphous materials such as organic biopolymers, the use of the XRD method can lead to a low-informative halo, the processing of which will not allow one to calculate the sizes of interplanar distances, crystalline regions, etc.
Thus, the use of combinations of XPS and XRD methods can certainly provide a significant amount of new data on the physicochemical state of the researched systems. However, because of the high complexity and novelty of these works, this will be the topic of our further work.
Thank you for your comments! You help us become better!
Round 2
Reviewer 2 Report
After the comprehensive feedback from reviewers, the paper has now been revised to a level where acceptance is possible. This said, there are still a few flaws that the authors need to take into account and apply appropriate corrections to (see attachment). Furthermore, the request for inclusion of a structural drawing representing the "substrate" 2-(4-acetamido-2-sulfanilimide) chitosan has not been rewarded and, thus, this demand has to be strongly repeated a second time!

Author Response
The structural formula of the matrix was added to the manuscript (see Scheme 1), and we also tried to take into account your other comments (new edits are highlighted in green, old ones are in yellow).
Thanks for your work! You help us get better!
Reviewer 3 Report
Dear Authors,
You addressed most of my remarks. However in my opinion the Introduction and Conclusion sections need your additional attention and I repeat my previous suggestions.
1) Introduction (Page 1-3)
- The main suggestion is to point out if the 2-(4-acetamido-2-sulfanilamide) chitosan has been used for the first time in this study. Or there were previous literature reports on this topic?
- The main goals of the study and the potential of such immobilized preparation (protease – chitosan derivative) should be more emphasized. I suggest putting 2-3 sentences about the synergistic effect of combining the antimicrobial properties of a chitosan-based matrix and the catalytic ability of tested enzymes to decompose the bacterial cell wall proteins.
2) Conclusions (Page 10)
- Authors are asked once again to give a comparison of their results with those obtained by other research groups or an explanation that they give such immobilized enzymes for the first time.
- Moreover, they are asked to give some information on how the immobilized biocatalysts they received could be applied in medicine as antibacterial systems.
- The future perspective of this research direction should be described as well.
I have an additional remark to the inserted paragraph (Page 2, lines 78 – 86):
In my opinion, this paragraph about carbon materials such as carbon nanotubes, graphene or its oxide is off top of the main purpose of the study. I would like to ask the Authors to replace this description with state-of-the-art on using chitosan and its derivatives as matrices for enzyme immobilization.

Author Response
1. Information on the use of chitosan as a matrix for protein immobilization has been added to the Introduction, as well as on the potential synergistic effect of practically significant properties of the obtained immobilized enzymes (new edits are highlighted in green, old edits are in yellow). We were unable to find any information on the use of the proposed matrix as a carrier for the enzyme.
2. Information about the prospects for the use of the received materials has been added to the Conclusion. The analysis of scientific information on the immobilization of bromelain, ficin and papain showed that for them the phenomenon of hyperactivation was not observed earlier. We have added this information to the Introduction.
3. We tried to take into account your comments and edited the paragraph about the use of various carriers for enzyme immobilization.
Thanks for your work! You help us get better!
Reviewer 4 Report
The authors improve their manuscript during the revision process. They responded to the majority of the comments.
For these reasons, I recommend the revised article article to be published in the journal.
Author Response
Thanks for your work! You help us get better!